# Cryptic Species Exist in *Vietnamella sinensis* Hsu, 1936 (Insecta: Ephemeroptera) from Studies of Complete Mitochondrial Genomes

**DOI:** 10.3390/insects13050412

**Published:** 2022-04-26

**Authors:** Yao Tong, Lian Wu, Sam Pedro Galilee Ayivi, Kenneth B. Storey, Yue Ma, Dan-Na Yu, Jia-Yong Zhang

**Affiliations:** 1College of Chemistry and Life Science, Zhejiang Normal University, Jinhua 321004, China; tyty9901@163.com (Y.T.); wulian@zjnu.edu.cn (L.W.); paydrov17@gmail.com (S.P.G.A.); ydn@zjnu.cn (D.-N.Y.); 2Department of Biology, Carleton University, Ottawa, ON K1S5B6, Canada; KenStorey@cunet.carleton; 3Key Lab of Wildlife Biotechnology, Conservation and Utilization of Zhejiang Province, Zhejiang Normal University, Jinhua 321004, China

**Keywords:** mitochondrial genome, cryptic species, phylogenetic relationship, divergence time

## Abstract

**Simple Summary:**

The family Vietnamellidae (Ephemeroptera) is one of the oldest insect families in the world. However, there are still controversies about the phylogenetic relationships among Vietnamellidae, Ephemerellidae, and Teloganodidae. The mitochondrial (mt) genome can be used to discuss phylogenetic relationships and cryptic species. We sequenced and annotated three complete mt genomes of *Vietnamella sinensis* from three different populations, identifying a cryptic species of *V. sinensis* and discuss the phylogenetic relationships of Vietnamellidae. Based on the genetic distance of the whole mt genomes, the phylogenetic relationship of three populations was uncovered and the divergence time of *V. sinensis* QY indicated that it was a cryptic species of *V. sinensis*.

**Abstract:**

Ephemeroptera (Insecta: Pterygota) are widely distributed all over the world with more than 3500 species. During the last decade, the phylogenetic relationships within Ephemeroptera have been a hot topic of research, especially regarding the phylogenetic relationships among Vietnamellidae. In this study, three mitochondrial genomes from three populations of *Vienamella sinensis* collected from Tonglu (*V. sinensis* TL), Chun’an (*V. sinensis* CN), and Qingyuan (*V. sinensis* QY) in Zhejiang Province, China were compared to discuss the potential existence of cryptic species. We also established their phylogenetic relationship by combining the mt genomes of 69 Ephemeroptera downloaded from NCBI. The mt genomes of *V. sinensis* TL, *V. sinensis* CN, and *V. sinensis* QY showed the same gene arrangement with lengths of 15,674 bp, 15,674 bp, and 15,610 bp, respectively. Comprehensive analyses of these three mt genomes revealed significant differences in mt genome organization, genetic distance, and divergence time. Our results showed that the specimens collected from Chun’an and Tonglu in Zhejiang Province, China belonged to *V. sinensis*, and the specimens collected from Qingyuan, Zhejiang Province, China were a cryptic species of *V. sinensis*. In maximum likelihood (ML) and Bayesian inference (BI) phylogenetic trees, the monophyly of the family Vietnamellidae was supported and Vietnamellidae has a close relationship with Ephemerellidae.

## 1. Introduction

The mitochondrial (mt) genome is the most extensively studied genomic system in insects, and is widely used to explore phylogenetic relationships due to its characteristics of maternal inheritance and high evolutionary rate. [1]. Insect mt genomes are usually double-stranded circular molecules of 14–20 kb in length, encoding 13 protein-coding genes (PCGs), 22 transfer RNAs (tRNAs), two ribosomal RNAs (rRNAs), and a control region (CR; AT-rich region) [2]. Many researchers have stimulated great interest in the insects’ mt genome including testing hypotheses about microevolution, mt gene expression, population structure analysis, phylogenetic relationships, and identification of cryptic species [1,3,4,5,6,7,8,9,10,11,12,13,14,15,16,17]. Since William Derham discovered the first cryptic species in 1718, these species are still a principal subject of several research areas and actually have a history of 300 years [18]. The study of cryptic species not only has great significance for the promotion of theories in related fields, but also is important for the quantifying biodiversity and conservation of species. Despite some controversy, most current studies define a cryptic species as one that is indistinguishable in morphology but significantly differentiated at the genetic level [19].

Ephemeroptera are generally referred to as mayflies, but have other aliases such as dayflies and fishflies [20]. Mayflies are an ancient lineage of insects, of which there are now 42 families, 400 genera, and 3500 species [21]. As early as 1988, Elliott et al. showed that Ephemeroptera play a significant role among freshwater fauna and can be used as one of the standards for water quality testing [22]. Ephemeroptera have a worldwide distribution, occurring on many continents, large islands, and archipelagos (except for Antarctica) [23,24]. Because Ephemeroptera are relatively primitive among Pterygota, considerable effort has been devoted to constructing the phylogenetic relationships within Ephemeroptera based on morphology [25,26,27], molecular evidence [28], and combined data [14,29]. The controversial points are mainly the following aspects: the phylogenetic relationship of Heptageniidae and Baetidae, and the phylogenetic relationship among Vietnamellidae, Teloganodidae, and Ephemerellidae [29,30,31].

The genus *Vietnamella* (Ephemeroptera: Vietnamellidae) was originally established by Tshernova in 1972, based on *Vietnamella thani* collected in Vietnam, and described the morphological structure of nymphs [32,33]. In 1997, McCafferty and Wang established that the subfamily Austremerellinae of the family Teloganodidae included the genus *Vietnamella* and *Austremerella* [34]. Then, this subfamily was elevated to family rank in 2000 and named Vietnamellidae [35]. In 2006, Jacobus and McCafferty moved *Austremerella* back to an indeterminate Austremerellidae and limited Vietnamellidae to include only one genus, *Vietnamella* [36]. In 2017, Hu et al. described *V. dabieshanensis* You and Su, 1987, *V. qingyuanensis* Zhou and Su, 1995, and *V. guadunensis* Zhou and Su, 1995 as junior synonyms of *V. sinensis* Hsu, 1936 [37]. As of 2021, there are six valid described *Vietnamella* species including *V. thani* Tshernova, 1972, *V. ornata* Tshernova, 1972, *V. sinensis*, *V. maculosa* Auychinda et al., 2020, *V. nanensis* Auchyinda et al., 2020, and *V. chebalingensis* Tong, 2020 [32,38,39,40].

The phylogenetic relationships among Vietnamellidae, Teloganodidae, and Ephemerellidae are controversial. In the early years, the most widely accepted early classification system was from McCafferty and Kluge, in which the phylogenetic relationship of Vietnamellidae was closer to Teloganodidae and more distant to Ephemerellidae [27,34]. In 2005, morphological data used by Ogden and Whiting suggested that Vietnamellidae was the sister clade to Teloganodidae, and the clade of (Vietnamellidae + Teloganodidae) formed a sister group to Ephemerellidae [28]. In 2009, Ogden et al. combined the molecular data (*sRNA*, *IrRNA*, *18S rDNA*, *28S rDNA*, and *H_3_* genes) and morphological data suggesting that Vietnamellidae, Teloganodidae, and Ephemerellidae were in a parallel relationship, and the phylogenetic relationships among them had not been effectively analyzed [29]. In recent years, Li et al. [41,42,43], Zhang et al. [44], Cai et al. [12], and Gao et al. [45] reconstructed the phylogeny of Ephemeroptera based on mt genomes, all of which supported the formation of sister groups between Ephemerellidae and Vietnamellidae.

In the present study, we sequenced the mt genomes from three populations of *V. sinensis* from Chun’an, Tonglu, and Qingyuan in Zhejiang Province, China to discuss the phylogenetic relationship of Vietnamellidae and the cryptic species of *V. sinensis*.

## 2. Materials and Methods

### 2.1. Sample Collection and Morphological Identification

The *Vietnamella* larvae were collected by D-frame aquatic insect net from streams of Tonglu (29°79′ N, 119°68′ E), Chun’an (24°41′ N, 120°52′ E), and Qingyuan (27°61′ N, 119°06′ E), Zhejiang Province, China in July 2019. The morphological structure of *Vietnamella* larvae including leg, head, antenna, maxilla, labium, hypopharynx, mandible, labrum and circus were dissected, observed, and photographed under an optical SMZ-1500 stereomicroscope (Nikon, Tokyo, Japan). After morphological identification, samples were stored in 100% ethanol at Zhang’s laboratory, College of Life Science and Chemistry, Zhejiang Normal University, China. According to the description of Hu et al. [37], the three populations of *Vietnamella* larvae were identified as *V. sinensis*.

### 2.2. DNA Extraction, PCR Amplification, and Sequencing

Total genomic DNA from hind-leg muscle tissue of each specimen was extracted using the Ezup Column Animal Genomic DNA Purification Kit (Sangon Biotech Company, Shanghai, China). The experimental design was approved by the Animal Research Ethics Committees of Zhejiang Normal University.

Several fragments were amplified using universal primers [6]. We then used Primer Premier 5.0 to design specific primers based on sequences previously obtained from universal primers [46]. The reaction conditions for normal PCR (product length <3000 bp) and long PCR (product length >3000 bp) were performed as described in Zhang et al. [47]. After we performed electrophoresis and gel purification of PCR products, all PCR products were sequenced at Sangon Biotech Company (Shanghai, China) using bidirectional sequencing.

### 2.3. Mitochondrial Genome Annotation and Sequence Analyses

Manual proofreading and splicing of all nucleotide fragments were conducted using SeqMan in the DNASTAR Package v.7.1 [48]. Annotation of tRNA genes and identification of their secondary structures were identified by the online website MITOS (http://mitos.bioinf.uni-leipzig.de/index.py) (accessed on 15 September 2021) [49]. Referring to the complete mt genomes of other Ephemeroptera species, we identified the two rRNA genes (12S and 16S rRNA) using the Clustal W program of Mega 7.0 [50] and aligned the amino acid sequences of PCGs. Mega 7.0 was also used to calculate the genetic distance. We calculated the AT content and relative synonymous codon usage rate (RSCU) of the three genomes [16] using PhyloSuite v.1.2.2 [51]. With reference to invertebrate genetic codes, the nucleic acid sequences of the 13 PCGs were translated into amino acid sequences [16,52]. The three mt genomes were deposited in GenBank with accession numbers OK265109–OK265111. We used CG View online server V 1.0 to draw maps of the mt genomes (http://cgview.ca/) (accessed on 20 October 2021) [53], and separately calculated the AT and GC skews according to the following formulae: AT skew = (A − T)/(A + T) and GC skew = (G − C)/(G + C) [54].

### 2.4. Phylogenetic Analyses

In order to further explore the phylogenetic relationship of Vietnamellidae within Ephemeroptera, we combined the three mt genomes in this study with 69 previously sequenced Ephemeroptera mt genomes (Appendix A) downloaded from NCBI including sequences from Ameletidae (1), Baetidae (4), Caenidae (5), Ephemerellidae (12), Ephemeridae (4), Heptageniidae (24), Isonychiidae (4), Leptophlebiinae (3), Polymitarcyidae (1), Potamanthidae (3), Siphlonuridae (5), Teloganodidae (2), and Vietnamellidae (3) [8,9,12,13,14,41,45,55,56,57,58,59,60,61,62].

Bayesian inference (BI) and maximum likelihood (ML) phylogenetic trees were constructed based on a dataset of the nucleotide sequences of 13 PCGs. Because the third codon was saturated, all phylogenetic analyses were performed using the first and second codons. In addition, two mt genomes of *Siphluriscus chinensis* (HQ875717, MF352165), the most primitive species of Ephemeroptera, was selected as the outgroup species [57,62]. Our test for the heterogeneity of nucleotide sequences between taxa used AliGROOVE with the default settings [63]. In this study, the nucleotide sequences of the 13 PCGs were used for DNA alignment, which was conducted in MAFFT v.7.475 [64]. Gblock 0.91b [65] and PartionFinder version 2.2.1 [66] were employed to select the conserved regions and partitions based on Bayesian information criterion (BIC). The partition schemes and best-fit models selected for each dataset are shown in Table 1. The GTR + I + G model was used to construct ML and BI analyses. The ML analysis was performed by RaxML 8.2.0 software with rapid inference using 1000 ultrafast repetitions [67]. The BI analysis was performed by MrBayes 3.2 [68] to 10 million generations and the mean standard deviation of Bayesian split frequencies was less than 0.01. The first 25% of generations was discarded as burn in. Tracer v1.7.1 [69] was used to detect the convergence to the stationary distribution of the chains. FigTree 1.4 was used to visualize the resulting trees [70].

### 2.5. Divergence Time Estimation

This study provided fossil calibration points to accurately estimate divergence time [71]. The use of fossil evidence to estimate the divergence time can deduce other taxa without fossils [72]. In total, we included four Ephemeroptera fossils for time-calibration to access the divergence time of Ephemeroptera. The first of the calibration points was the divergence time in Siphlonuridae (159.00–160.60 Mya, an average of 159.80 Mya) [73,74], which was found for new middle Jurassic mayflies from inner Mongolia. The second calibration point was the divergence time in Atalophlebiinae of Leptophlebiidae (15.00–20.00 Mya, 17.50 Mya average) [75,76]. We used this calibration point for *Choroterpides apiculate*, which belongs to the subfamily Atalophlebiinae. The third calibration point was the divergence time in the genus *Ephemerella* of Ephemerellidae (41.30–47.80 Mya, 44.55 Mya average) [77]. The final calibration point was the divergence time in Vietnamellidae (98.17–99.41 Mya, 98.79 Mya average) [78], and this fossil was the first fossil record of Vietnamellidae from Mid-Cretaceous Burmese amber. The root age was set to 239 million years ago (the oldest age of Ephemeroptera) [79]. All fossil calibrations used the uniform model.

Based on the topology of maximum likelihood (ML) phylogenetic tree, MCMCTree [80] in the PAML v4.8 package [81] was performed to infer divergence time in Ephemeroptera. In conducting the analysis, we calculated the transition/transversion rate ratio and branch lengths first. When running MCMC, the parameters of the algorithm were set as: burn-in period = 1,000,000, sample frequency = 1000, and number of samples = 10,000. Since MCMC runs need to check convergence, they were run at least twice from different starting points. To check the convergence, we cut the time values from two output files and pasted them into an Excel spreadsheet, then used Excel to draw a scatter diagram of the time set for the first and second runs. The points should be very close to a straight line (the *x* = *y* line) [80]. Tracer v1.7.1 [69] was used to confirm the adequate mixing of the MCMC chains of the run and to check that the marginal posterior distribution value (ESS, effective sample size) were greater than 200. Finally, FigTree v1.4 was used to visualize the divergence time of every branch [70].

## 3. Results

### 3.1. Mitochondrial Genome Composition

We annotated and uploaded the complete mt genome data of *V. sinensis* TL (OK265109), *V. sinensis* CN (OK265111), and *V. sinensis* QY (OK265110) into the GenBank database. These three mt genomes all showed double circular DNA molecules with lengths of 15,674 bp, 15,674 bp, and 15,610 bp, respectively (Figure 1A,B). The intergenic nucleotides (IGNs) of the three species ranged from 1 to 41 bp (Appendix A) and the gene order of the three mt genomes were the same as those of typical insects, with a total 37 genes and an A + T rich region including 22 tRNA genes, two rRNA genes, and 13 PCGs. Of these 37 genes, 23 genes were located on the heavy (H) strand and the remaining 14 genes were on the light (L) strand (Appendix A). After subsequent analysis, *V. sinensis* CN and *V. sinensis* TL had the same length and similar characteristics, so *V. sinensis* CN and *V. sinensis* TL were collectively referred to as *V. sinensis* CN/TL in this paper. The nucleotide composition of the *V. sinensis* CN/TL mt genome was A = 32.3%, T = 38.2%, C = 17.8%, and G = 11.7%, and was very similar to *V. sinensis* QY, which was A = 31.9%, T = 37.6%, C = 18.4%, and G = 12.1%. There were strong A + T biases in both *V. sinensis* CN/TL and *V. sinensis* QY, 70.5% and 69.5%, respectively. According to the skew statistics, the AT skew was positive, whereas the GC skew was negative (Table 2).

In both *V. sinensis* CN/TL and *V. sinensis* QY mt genomes nine genes (ND2, COI, COII, COIII, ATP6, ATP8, ND3, ND6 and Cyt b) were located on the heavy strand (H-strand), whereas the others (ND4, ND4L, ND5, and ND1) were on the light strand (L-strand) (Appendix A). Both *V. sinensis* CN/TL and *V. sinensis* QY had the same PCG lengths of 11,226 bp (Table 2). Among the three sequenced mt genomes, 13 PCGs used the typical mitochondrial start codon ATN, and 10 PCGs used typical stop codons TAA and TAG. An incomplete codon T was used for the other three PCGs (COII, ND5, and Cyt b). G-C skew values were all negative in PCGs (+), but positive in PCGs (−). Unlike G-C skew values, A-T skew values were all negative in both PCGs (−) and PCGs (+) (Table 2).

The relative synonymous codon usage (RSCU) of the three mt genomes of *V. sinensis* was calculated (Figure 2, Appendix A). The most commonly codons in the PCGs of *V. sinensis* CN/TL and *V. sinensis* QY were UUU (Phe), UUA (Leu), and AUU (Ile), with a frequency of >230 times.

The mt genomes of *V. sinensis* CN/TL and *V. sinensis* QY possessed 22 tRNA genes. Among these genes, most tRNAs (14) were encoded on the heavy strand (H-strand), whereas eight tRNAs were encoded on the light strand (L-strand) (Appendix A). The tRNAs exhibited the classic cloverleaf secondary structure. The total tRNA sizes of *V. sinensis* CN/TL and *V. sinensis* QY were 1429 bp and 1433 bp, respectively. In both cases, the smallest tRNA was trnC with a length of 61 bp, whereas the largest tRNA was trnY with a length of 70 bp. There were differences between the tRNA secondary structures of these three mt genomes (Figure 3). The 16S rRNA gene was located between trnL and trnV with lengths of 1223 bp and 1221 bp in *V. sinensis* CN/TL and *V. sinensis* QY, respectively (Appendix A). The 12S rRNA gene was located between trnV and the CR with lengths of 792 bp and 790 bp in *V. sinensis* CN/TL and *V. sinensis* QY, respectively. The AT content of rRNA in these three mt genomes was between 74.1% and 74.3%, and the nucleotide skew was all positive for AT and GC (Table 2).

The control region was located between the 12S rRNA and trnI genes. The length and distribution of the control region in these mt genomes of *Vietnamella* were relatively conservative. The length of the A + T—rich region of *V. sinensis* CN/TL and *V. sinensis* QY mt genomes was 1015 bp and 911 bp, respectively.

### 3.2. Analysis of Genetic Distance

Based on the difference in organization and composition of the mt genome, the complete mt genomes of *V. sinensis* CN, *V. sinensis* TL, and *V. sinensis* QY was used to explore their genetic distance. Within Vietnamellidae, the genetic distance between all known species ranged from 0.1% to 21.1%, with an average of 15.58% (Table 3). The genetic distance of the two specimens from the Chun’an and Tonglu populations were similar (0.1%), and the two collection locations were about 87 km apart, suggesting that these two specimens belong to the same species. The genetic distance between *V. sinensis* QY and the two other species was 14.8% for *V. sinensis* CN and 14.9% for *V. sinensis* TL, respectively. Comparison of the current data with other previous reports for *V. sinensis* was also conducted. The calculated genetic distance between *V. sinensis* CN/TL and *V.* sp. MT-2014 (KM244655) versus between *V. sinensis* CN/TL and *V.* sp. JZ-2021 (MF352146) was 20.9% and 18.3%, respectively. However, the genetic distance between *V. sinensis* CN/TL and *V. sinensis* (HM067837) was 5.8%. The genetic difference between *V. sinensis* QY and *V. sinensis* (HM067837) was 14%, reaching the level of species. Hence, the data indicate that *V. sinensis* QY is a cryptic species of *V. sinensis* according to genetic distance.

### 3.3. Heterogeneous Sequences Divergence and Phylogenetic Analyses

The obtained AliGROOVE matrixes helped to detect the pairwise comparisons of nucleotide datasets among all taxon comparisons that had positive similarity scores (Figure 4). Average similarity scores analyzed between sequences are represented by colored squares, and all colored squares ranged from −1 (indicating large differences in ratios to the rest of the dataset) to +1 (indicating ratios match in all other comparisons). The individual matrixes in the results revealed the degree of heterogeneity in the PCG12 matrix dataset (Figure 4). The pairwise sequence comparisons between the dataset showed a high degree of similarity, whereas the family Baetidae species showed high heterogeneity. This heterogeneity may be related to phylogenetic long-branch attraction (LBA) (Figure 5).

This study used the PCG12 matrix dataset of 72 Ephemeroptera species to analyze phylogenetic relationships, and the results showed that the monophyly of most families were supported in the phylogenetic trees. However, the presence of only one species in Ameletidae, Polymitarcyidae, and Teloganodidae limited a discussion of their monophyletic analysis (Figure 5). According to the results of phylogenetic topologies, Isonychiidae was a sister group to the other families within ingroups of Ephemeroptera. After that, Ameletidae and Siphlonuridae were found to be a sister group. Heptageniidae was a sister clade for the remaining Ephemeroptera (Heptageniidae + (((Leptophlebiidae + (Caenidae + (Teloganodidae + Baetidae))) + (Ephemerellidae + Vietnamellidae)) + (Potamanthidae + (Polymitarcyidae + Ephemeridae))). The clade of (Vietnamellidae + Ephemerellidae) was a sister clade to the clade of (Leptophlebiidae + (Caenidae + (Baetidae + Teloganodidae))).

Long-branch attraction (LBA) was observed in Teloganodidae (*Teloganodidae* sp. MT-2014) and Baetidae (*Baetis* sp. PC-2010, *Baetis* sp. 2 ZY-2019, *Takobia yixiani* and *Cloeon dipterum*) in both BI and ML analyses. The phylogenetic relationships between BI and ML showed roughly identical topologies, except for differences in the position of *Vietnamella* sp. JZ-2021 (MF352146) within *Vietnamella*. The ML tree showed the phylogenetic relationship of (((*V. sinensis* CN + *V. sinensis* TL) + *V. sinensis*) + *V. sinensis* QY) + (*V.* sp. MT-2014 + *V.* sp. JZ-2021). In contrast, the BI tree showed a phylogenetic relationship of (((((*V. sinensis* CN + *V. sinensis* TL) + *V. sinensis*) + *V. sinensis* QY) + *V.* sp. JZ-2021) + *V.* sp. MT-2014). Concentrating on the phylogenetic relationship of Vietnamellidae, Vietnamellidae was a sister clade to Ephemerellidae and has a distant relationship with Teloganellidae. In both BI and ML trees, the clade of *V. sinensis* CN and *V. sinensis* TL was sister clade to *V. sinensis*, and then were clustered together with *V. sinensis* QY. We realized that *V. sinensis* QY was a distant sister clade to *V. sinensis* CN and *V. sinensis* TL. In general, in accordance with previous phylogenetic studies based on morphological characteristics and molecular data, the monophyly of all Ephemeroptera families in this study was supported, except for Siphluriscidae. However, the monophyly of families (Ameletide, Teloganodidae and Polymitarcyidae) with only one species needs to be further explored.

### 3.4. Divergence Time Estimation

In this study, the PCG12 dataset was selected for use in divergence time estimation. The results of our divergence time analysis are shown Figure 6 and Table 4. Analysis of divergence time revealed that the diversification of Ephemeroptera occurred about 196.91 million years ago (Mya) [95% HPD (highest posterior densities), 171.18–236.55 Mya] (Figure 6). Isonychiidae originated in the Jurassic [187.82 Mya; 95% HPD, 168.38–223.63 Mya], and was a sister group to the other families within Ephemeroptera excluding Siphluriscidae. The most recent common ancestor (MRCA) of Ameletidae and Siphlonuridae also diverged in the Jurassic [159.99 Mya; 95% HPD, 159.00–161.00 Mya]. The MRCA of Heptageniidae was estimated to be at 173.64 Mya [95% HPD, 155.70–206.74 Mya]. Our results indicate that the eight currently accepted families of Ephemeridae, Polymitarcyidae, Potamanthidae, Vietnamellidae, Ephemerellidae, Caneidae, Baetidae, and Teloganodidae have similar divergence times from within the Cretaceous.

We estimated that Vietnamellidae appeared during the Cretaceous (98.50 Mya; 95% HPD, 98.00–99.00 Mya), which is supported by the first fossil record of the mayfly family Vietnamellidae from Burmese amber [78]. The MRCA of the three species in the current study and *V. sinensis* was estimated to be at 10.88 Mya [95% HPD, 5.11–19.87 Mya]. After *V. sinensis* QY was separated from the branches, then *Vietnamella sinensis* was separated again from *V. sinensis* CN and *V. sinensis* TL at around 3.95 Mya [95% HPD, 1.37–8.87 Mya]. Our analyses recovered a divergence between *V. sinensis* CN and *V. sinensis* TL, which is estimated to have occurred during the Neogene (0.12 Mya; 95% HPD, 0.001–0.42 Mya).

## 4. Discussion

### 4.1. Comparison of Mitochondrial Genome Composition

In 2017, Hu et al. described *V. dabieshanensis*, *V. qingyuanensis*, and *V. guadunensis* as junior synonyms of *V. sinensis* [37]. Therefore, in subsequent comparisons, *V. dabieshanensis* (HM067837) will be referred to as *V. sinensis*. As of January 2022, the National Center for Biotechnology Information (NCBI) has released three mt genomes of the *Vietnamella*: *V. sinensis* (HM067837, 15,761 bp), *Vietnamella.* sp. MT-2014 (KM244655, 15,043 bp), and *Vietnamella.* sp. JZ-2021 (MF352146, 15,043 bp). The three mt genomes of this study were shorter than the complete mt genome of *V. sinensis*, and the size differences of these mt genomes was mainly caused by the size of the intergenic nucleotides (IGNs) and the CR (Appendix A). The intergenic nucleotides (IGNs) of *V. sinensis* CN/TL and *V. sinensis* QY ranged from 1 to 41 bp (Appendix A), and these were identical with *V. sinensis* (HM067837). In addition, we analyzed the sizes and nucleotide compositions of *V. sinensis* (HM067837). The A – T content of the whole mt genome, PCGs, tRNA, and rRNA were calculated, and the results ranged from 66.4% to 71.8% (PCGs), 71.3% to 74.9% (tRNA), and 74.1% to 74.3% (rRNA), respectively (Table 2). There were strong A + T biases in both *V. sinensis* CN/TL and *V. sinensis* QY, 70.5% and 69.5%, respectively, and together, these were similar to *V. sinensis* (HM067837) (70.7%).

The sizes of PCGs in the three mt genomes (11,226 bp) were similar to *V. sinensis* (11,121 bp). We found that the start and stop codons of the 13 PCGs in both *V. sinensis* CN/TL and *V. sinensis* QY were the same as in *V. sinensis* (HM067837). Among the 13 PCGs, 10 PCGs used typical stop codons and three PCGs used incomplete stop codons. Incomplete stop codons are common in metazoan mt genomes [82]. We also observed that the AT content of PCGs (−) in the three sequenced mt genomes and *V. sinensis* (HM067837) was greater than that in PCGs (+) (Table 2). The balance between mutation, selection pressure, and genetic drift can lead to codon usage bias, so codon usage analysis is important for understanding genome evolution [83,84]. Due to the results that AT mutation bias has effects on codon usage, we found that codons with the third nucleotide being G or C in Appendix A were rarely used [85,86]. It can be seen from Appendix A that the codon count and RSCU of *V. sinensis* CN/TL and *V. sinensis* QY were relatively similar.

With few exceptions, most metazoan mt genomes contain 22 tRNA genes including two trnL (UUR and CUN) and two trnS (AGN and UCN) [87]. Among the 22 tRNA genes of the three mt genomes, the secondary structure of most tRNA genes was the normal cloverleaf model, except for trnI (*V. sinensis* CN/TL and *V. sinensis*), trnM (*V. sinensis* CN/TL and *V. sinensis*), and trnP (*V. sinensis* CN/TL, *V. sinensis*, and *V. sinensis* QY), which lacks the TΨC loops. Furthermore, trnS1 (*V. sinensis* CN/TL, *V. sinensis,* and *V. sinensis* QY) had lost the dihydrouridine (DHC) arm (Figure 3). A lack of TΨC loops or DHC arms can be found in other Ephemeroptera [57], and their translational activity was lower than the normal structures [88]. Quite a few mismatched pairs were found among the three species and *V. sinensis* (HM067837), and the specific mismatches are shown in Figure 3. It can be seen from the graph that the similarity of secondary structures of the tRNA genes between *V. sinensis* CN/TL and *V. sinensis* (HM067837) was higher than that of *V. sinensis* QY and *V. sinensis* (HM067837).

### 4.2. Phylogenetic Analyses of Vietnamellidae within Ephemeroptera

In recent years, the higher-level phylogenetic relationships within Ephemeroptera have been widely debated [8,26,27,34,89]. In most cases, *Siphluriscus chinensis* (HQ875717, MF352165) was primitively diverged from other mayflies within Ephemeroptera [57]. Vietnamellidae and Teloganodidae were in parallel relationship, and had a distant relationship with Ephemerellidae [27,28]. In our study, the family Vietnamellidae was strongly monophyletic in our topologies (Figure 5). Both maximum likelihood and Bayesian phylogenetic trees showed that Vietnamellidae was closely related to Ephemerellidae and had a distant relationship with Teloganellidae, which was consistent with the results of Cai et al. [12], Gao et al. [45], and Rutschmann et al. [90]. Over the last few years, many scholars did not include sequences of Teloganodidae when constructing the phylogenetic tree within Ephemeroptera, so the sister clade of Vietnamellidae and Ephemerellidae were still supported [41,42,43,44].

### 4.3. Identification of Cryptic Species

The intraspecific genetic distances of these three species varied between 0.1% (*V. sinensis* CN—*V. sinensis* TL) and 14.9% (*V. sinensis* CN/TL—*V. sinensis* QY), whereas the interspecific genetic distances with other *Vietnamella* species were very high, ranging from 18.3% (*V. sinensis* CN/TL—*V.* sp. JZ-2021) to 21.1% (*V. sinensis* QY—*V.* sp. MT-2014) (Table 3). The difference in genetic distance between *V. sinensis* CN/TL and *V. sinensis* (HM067837) published in the NCBI was 5.8%, which is between the 1% and 7% of typical insect reports [91]. However, the difference between *V. sinensis* QY and *V. sinensis* (HM067837) was 14%. Hence, we consider *V. sinensis* QY to be a cryptic species of *V. sinensis* (HM067837). Williams found that the genetic distance of *Baetis rhodani* in different geographic locations was 8–19% at the molecular level, and judged that some populations were cryptic species [92]. In this study, the genetic distance of *V. sinensis* QY reached 14%, which supports the conclusion that *V. sinensis* QY is a cryptic species of *V. sinensis* (HM067837).

Within Vietnamellidae, *V. sinensis* (HM067837) was a sister group to *V. sinensis* CN and *V. sinensis* TL, according to the phylogenetic topologies (Figure 5). *V. sinensis* QY was the earliest divergent lineage (10.88 Mya) from *V. sinensis* (HM067837) and was still quite closely relatively related (Figure 6). In this study, we estimated Vietnamellidae to have appeared during the Cretaceous period. After *V. sinensis* QY was separated from the main branches of *V. sinensis* at around 10.88 Mya, then *V. sinensis* was separated again from *V. sinensis* CN and *V. sinensis* TL at around 3.95 Mya, and the divergence time was far from *V. sinensis* QY. On the whole, not only the genetic distance of mt genomes, but also the phylogenetic analysis and divergence time of the three populations of *Vietnamella*, suggest that *V. sinensis* QY was a cryptic species of *V. sinensis*.

## 5. Conclusions

In this study, the complete mt genome sequences of *V. sinensis* CN, *V. sinensis* TL, and *V. sinensis* QY were successfully determined. The three *Vietnamella* mt genomes showed similar gene arrangements to *V. sinensis* (HM067837). BI and ML analyses indicated roughly identical topology, except for the position of *Vietnamella* sp. JZ-2021 (MF352146) within *Vietnamella*. Furthermore, our study showed that Vietnamellidae was the sister clade to Ephemerellidae, but Teloganellidae was far from Vietnamellidae and Ephemerellidae. Analysis of divergence time revealed that the diversification of Ephemeroptera occurred about 196.91 million years ago. Divergence times in most families suggested that most diversity arose during the Mesozoic era and Vietnamellidae appeared during the Cretaceous (98.50 Mya).

Overall, comprehensive comparative analysis of the mt genomes of *V. sinensis* CN, *V. sinensis* TL, and *V. sinensis* QY revealed that they differed significantly in various aspects such as genome composition, genetic distance, phylogenetic analyses, and divergence time estimation. The genetic distance between *V. sinensis* QY and *V. sinensis* (HM067837) reached 14%, which was much higher than that of *V. sinensis* CN, *V. sinensis* TL, and *V. sinensis* (HM067837), of 5.8%. In the analysis of phylogeny and divergence time estimation, *V. sinensis* QY first separated from *V. sinensis* (HM067837), *V. sinensis* CN, and *V. sinensis* TL about 10.88 Mya, and then *V. sinensis* (HM067837) separated from *V. sinensis* CN and *V. sinensis* TL at about 3.95 Mya. The results of this study provide evidence for the existence of cryptic species. The specific characteristics of mt genomes may be used as accessible and powerful molecular markers in the identification of cryptic species. Further research priorities can conduct additional studies of the mt genomes of Teloganodidae and Baetidae to explore long-branch attraction within Ephemeroptera.

## Figures and Tables

**Figure 1 insects-13-00412-f001:**
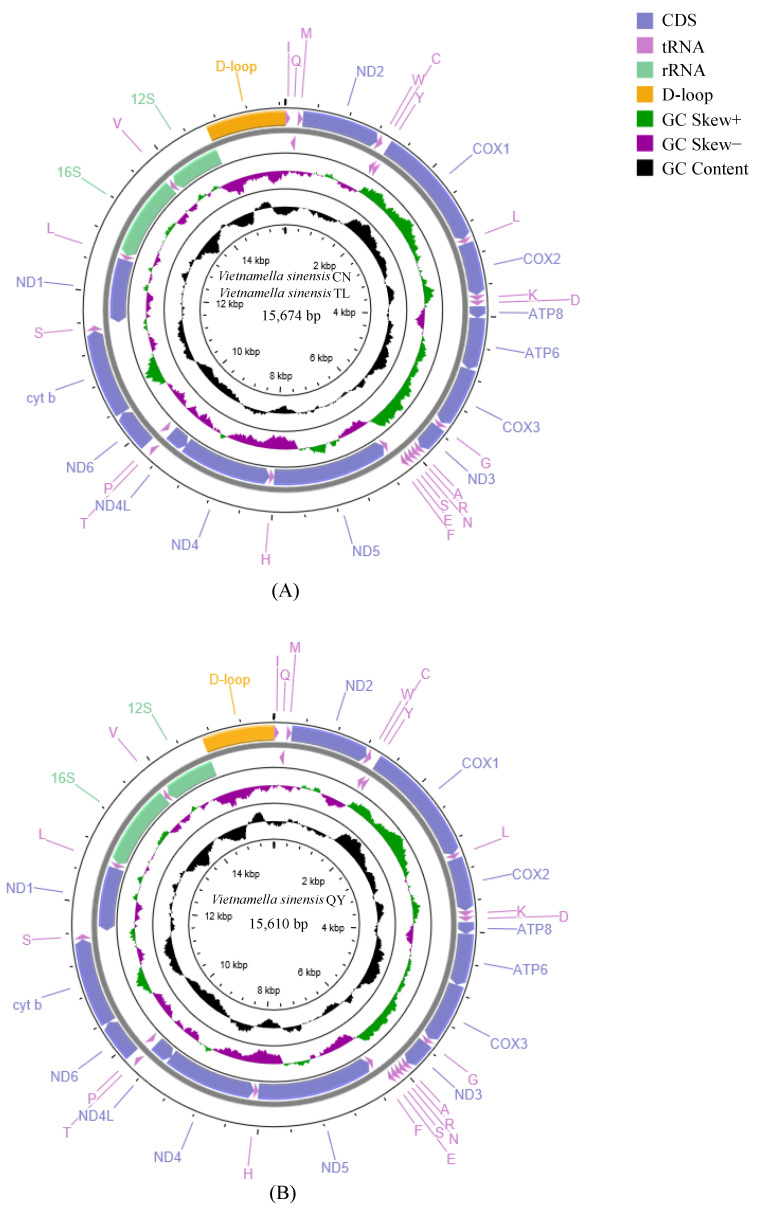
Circular visualization and organization of the complete mt genome of *V. sinensis* CN + *V. sinensis* TL (**A**) and *V. sinensis* QY (**B**). External genes on the circle are encoded by the positive strand (5′→3′) and internal genes are encoded by the negative strand (3′→5′). The second circle shows the GC skew and the third shows the GC content. GC content and GC skew are plotted as the deviation from the average value of the entire sequence. Other items are defined on the figure.

**Figure 2 insects-13-00412-f002:**
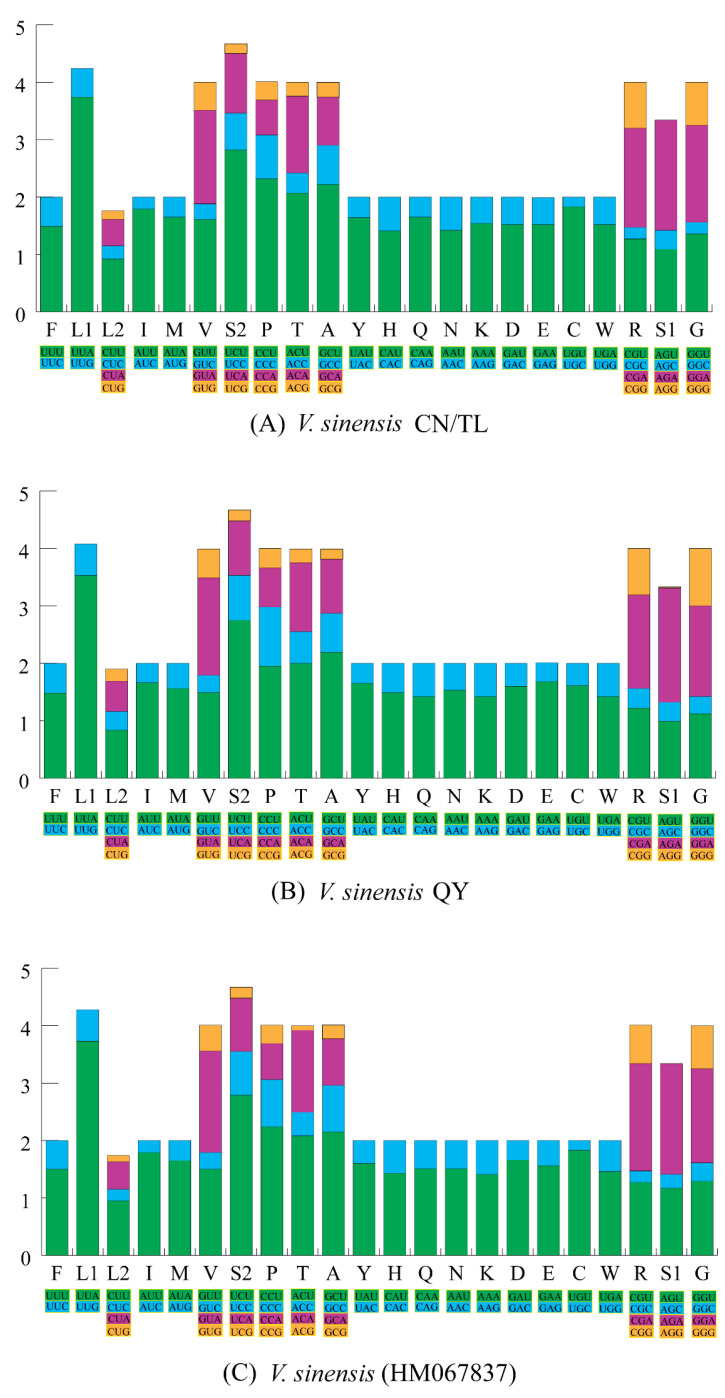
The relative synonymous codon usage (RSCU) of the mt genome in *V. sinensis* CN/TL (**A**), *V. sinensis* QY (**B**), and *V. sinensis* (HM067837) (**C**).

**Figure 3 insects-13-00412-f003:**
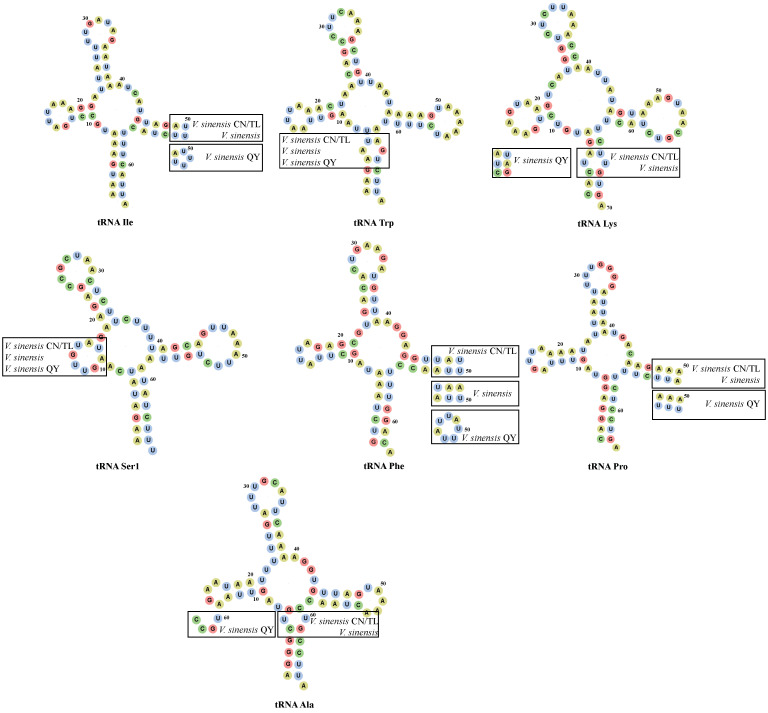
Inferred different secondary structures of the tRNA genes of *V. sinensis* CN/TL, *V. sinensis* QY, and *V. sinensis* (HM067837).

**Figure 4 insects-13-00412-f004:**
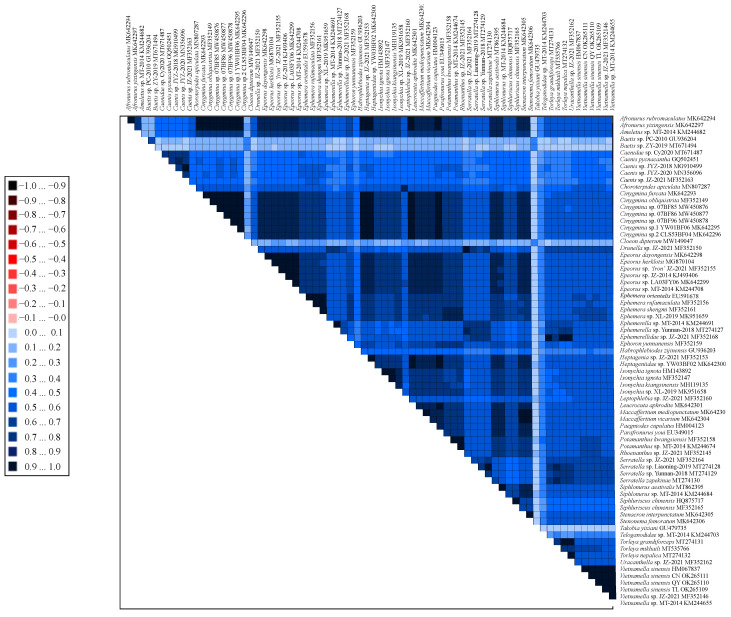
Heterogeneous sequence divergence within two datasets of PCGs of 72 Ephemeroptera mt genomes for the PCGs matrix datasets including three codon positions of PCGs.

**Figure 5 insects-13-00412-f005:**
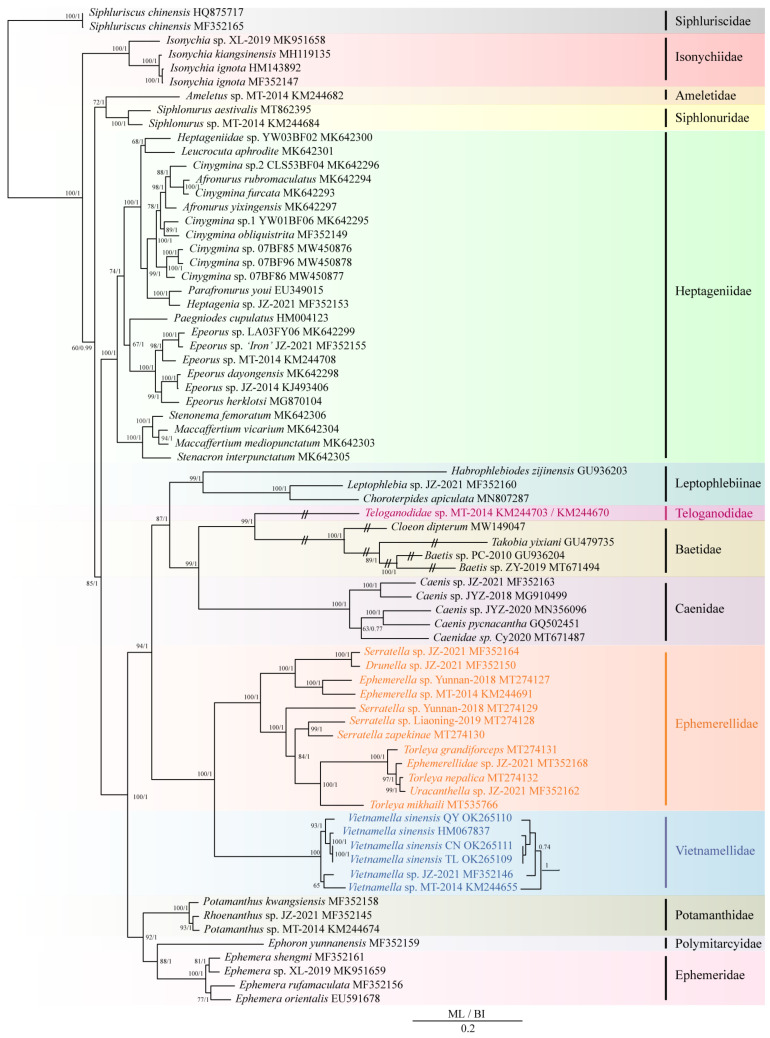
Phylogenetic tree of the relationships among 72 species of Ephemeroptera according to the nucleotide dataset of the 13 mt PCGs. *Siphluriscus chinensis* (HQ875717, MF352165) was used as the outgroup. The numbers above branches specify bootstrap percentages from ML (**left**) and posterior probabilities as determined from BI (**right**). The GenBank accession numbers of all species are shown in the figure. Long-branch attractions of Baetidae and Teloganodidae have been cut for aesthetics.

**Figure 6 insects-13-00412-f006:**
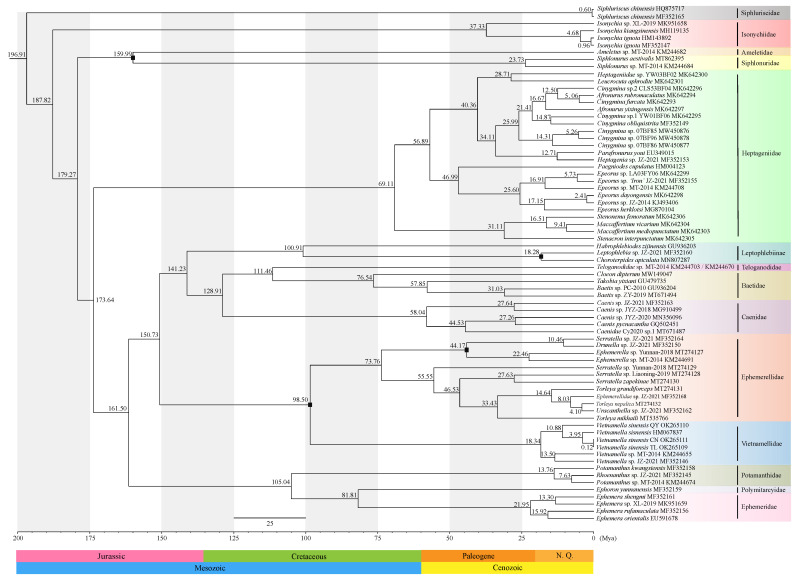
Evolutionary timescale for the Ephemeroptera inferred from the PCGs dataset based on phylogenetic analyses using four fossil calibration points. Each fossil calibration point is marked with a black dot on the figure. Median ages on the chronogram are provided above nodes. A geological time scale is shown at the bottom.

**Table 1 insects-13-00412-t001:** The partition schemes and best-fitting models selected. The complete names of all abbreviations are as follows: pos1: the first codon; pos2: the second codon; GTR: general time reversible; I: proportion of invariable sites; G: gamma distribution.

Nucleotide Sequence Alignments		
Subset	Subset Partitions	Best Model
Partition 1	COII_pos1, COIII_pos1, Cyt b_pos1, ATP6_pos1	GTR + I + G
Partition 2	COI_pos2, Cyt b_pos2, COII_pos2, ATP6_pos2, COIII_pos2	GTR + I + G
Partition 3	ND3_pos1, ND6_pos1, ATP8_pos1, ND2_pos1	GTR + I + G
Partition 4	ND6_pos2, ATP8_pos2, ND2_pos2, ND3_pos2, ND4L_pos2	GTR + I + G
Partition 5	COI_pos1	GTR + I + G
Partition 6	ND4L _pos1, ND1_pos1, ND4_pos1, ND5_pos1	GTR + I + G
Partition 7	ND1_pos2, ND5_pos2, ND4_pos2	GTR + I + G

**Table 2 insects-13-00412-t002:** Base composition of the mt genomes of the *V. sinensis* CN/TL, *V. sinensis* QY, and *V. sinensis* (HM067837).

Region	Strand	*V. sinensis* CN/TL				*V. sinensis* QY				*V. sinensis* (HM067837)			
		Length (bp)	AT%	AT Skew	GC Skew	Length (bp)	AT%	AT Skew	GC Skew	Length (bp)	AT%	AT Skew	GC Skew
Whole genome		15,674	70.5	−0.083	−0.208	15,610	69.5	−0.083	−0.207	15,761	70.7	−0.092	−0.197
PCGs	+	6915	67.9	−0.207	−0.157	6915	66.4	−0.208	−0.160	6915	67.7	−0.214	−0.153
	−	4311	71.6	−0.147	0.289	4311	70.9	−0.138	0.289	4308	71.8	−0.144	0.286
tRNA	+	910	71.3	−0.005	0.034	912	71.5	−0.009	0.031	915	71.3	0.002	0.031
	−	519	73.6	0.031	0.314	521	74.9	0.046	0.298	520	74.8	0.059	0.313
rRNA	−	2015	74.3	0.106	0.216	2011	74.2	0.111	0.215	2044	74.1	0.106	0.214

**Table 3 insects-13-00412-t003:** The genetic distance of the complete mt genomes within Vietnamellidae.

	Species	1	2	3	4	5
1	*V. sinensis* HM067837					
2	*V.* sp. MT-2014 KM244655	0.202				
3	*V.* sp. JZ-2021 MF352146	0.187	0.210			
4	*V. sinensis* CN OK265111	0.058	0.209	0.183		
5	*V. sinensis* TL OK265109	0.058	0.209	0.183	0.001	
6	*V. sinensis* QY OK265110	0.140	0.211	0.190	0.148	0.149

**Table 4 insects-13-00412-t004:** Divergence times for nodes/clades in the Ephemeroptera based on the mt genome. All the estimates are represented in millions of years ago (Mya). “&” represents the relationship between two branches.

Nodes/Clades	Mean Divergence Time (Mya)	95% HPD Range (Mya)
Ephemeridae & Polymitarcyidae	81.81	33.19~142.93
(Ephemeridae + Polymitarcyidae) & Potamanthidae	105.04	43.32~164.57
Vietnamellidae & Ephemerellidae	98.50	98.00~99.00
Teloganodidae & Baetidae	111.46	84.47~142.94
(Teloganodidae + Baetidae) & Caenidae	128.91	102.11~162.40
(Teloganodidae + (Baetidae + Caenidae)) & Leptophlebiinae	141.23	115.39~174.06
((Teloganodidae + (Baetidae + Caenidae)) + Leptophlebiinae) & (Vietnamellidae + Ephemerellidae)	150.73	126.64~183.03
(((Teloganodidae + (Baetidae + Caenidae)) + Leptophlebiinae) + (Vietnamellidae + Ephemerellidae)) & ((Ephemeridae + Polymitarcyidae) + Potamanthidae)	161.50	139.21~193.76
((((Teloganodidae + (Baetidae + Caenidae)) + Leptophlebiinae) + (Vietnamellidae + Ephemerellidae)) + ((Ephemeridae + Polymitarcyidae) + Potamanthidae)) & Heptageniidae	173.64	155.70~206.74
Siphlonuridae & Ameletidae	159.99	159.00~161.00
(((((Teloganodidae + (Baetidae + Caenidae)) + Leptophlebiinae) + (Vietnamellidae + Ephemerellidae)) + ((Ephemeridae + Polymitarcyidae) + Potamanthidae)) + Heptageniidae) & (Siphlonuridae + Ameletidae)	179.27	163.65~213.04
((((((Teloganodidae + (Baetidae + Caenidae)) + Leptophlebiinae) + (Vietnamellidae + Ephemerellidae)) + ((Ephemeridae + Polymitarcyidae) + Potamanthidae)) + Heptageniidae) + (Siphlonuridae + Ameletidae)) & Isonychiidae	187.82	168.38~223.63
(((((((Teloganodidae + (Baetidae + Caenidae)) + Leptophlebiinae) + (Vietnamellidae + Ephemerellidae)) + ((Ephemeridae + Polymitarcyidae) + Potamanthidae)) + Heptageniidae) + (Siphlonuridae + Ameletidae)) + Isonychiidae) & Siphluriscidae	196.91	171.18~236.55

## Data Availability

The data supporting the findings of this study are openly available from the National Center for Biotechnology Information at https://www.ncbi.nlm.nih.gov (accessed on 24 September 2021). Accession numbers are: OK265109, OK265110, and OK265111.

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
