# Peer review of "Cryptic Species Exist in Vietnamella sinensis Hsu, 1936 (Insecta: Ephemeroptera) from Studies of Complete Mitochondrial Genomes"

_insects, 2022, doi:10.3390/insects13050412_

Round 1
Reviewer 1 Report
Abstract
L38-39 Please, write ML and BL entirely
Introduction
Please include some information that make mitochondrial genomes appropriate for studying phylogenetic relationships
Materials and methods
Please provide all the parameters of softwares used during your analyses.
Please, explain how did you obtain the inferred different secondary structures (Figure 3).
L101 Please specify how did you collect your larvae
L106 Please specify any morphological keys or websites or resources used during your morphological identification
L111 What do mean by muscle tissue? Please specify it or them
L117 Please, give the exact size of your PCR product.
Please describe your electrophoresis and any treatment done on your PCR product before sequencing
Table 1 : Please, provide the complete names of all abbreviations.
Results
Table 2 : V. sinensis (HM067837) data are not visible
L407-408 That sentence sounds incomplete, either you complete it or you remove it.
L458-L511
Each figure should have its legend. You don't need to have a separate section for legend or notes. Please move them and add your legends and notes near of each figure or table.
Author Response
Reviewer #1:
- Abstract
(1) L38-39 Please, write ML and BI entirely
Response: Thanks for your suggestion. Added.
- Introduction
(1) Please include some information that make mitochondrial genomes appropriate for studying phylogenetic relationships
Response: We have added the characteristics of mitochondrial genomes appropriate for studying phylogenetic relationships in L48-49.
- Materials and methods
(1) Please provide all the parameters of softwares used during your analyses.
Response: The parameters of the software have been written in the article (such as L164-165), and the default parameters are used if not specified (such as L155-156). So, we provided some parameters of software.
(2) Please, explain how did you obtain the inferred different secondary structures (Figure 3).
Response: First, annotation of tRNA genes and identification differences of their secondary structures were identified by the online website MITOS (http://mitos.bioinf.uni-leipzig.de/index.py). Then we used ViennaRNA Web Services (http://rna.tbi.univie.ac.at/forna/) to beautify the difference in secondary structure.
(3) L101 Please specify how did you collect your larvae
Response: We have supplemented the method for collecting larvae at L106.
(4) L106 Please specify any morphological keys or websites or resources used during your morphological identification
Response: In L113-114 we have indicated that the morphological identification was performed according to the article by Hu et al, 2017.
(5) L111 What do mean by muscle tissue? Please specify it or them
Response: We have completed this sentence at L116.
(6) L117 Please, give the exact size of your PCR product.
Response: Since the length of normal PCR products were in the range of several hundred to three thousand (475 bp-2342 bp), and the length of long PCR were more than three thousand (3056 bp-3785 bp). We think it is not necessary to give the exact size in the PCR product in the paper, because some papers were shown those, eg. Zhang et al. 2008.
(7) Please describe your electrophoresis and any treatment done on your PCR product before sequencing
Response: Added. “After we performed electrophoresis and gel purification of PCR products, ”
(8) Table 1: Please, provide the complete names of all abbreviations.
Response: At Table 1, we made a supplementary explanation for each abbreviation.
- Results
(1) Table 2: V. sinensis (HM067837) data are not visible
Response: We made adjustments in Table 2.
(2) L407-408 That sentence sounds incomplete, either you complete it or you remove it.
Response: Thanks for your suggestion, we deleted this sentence.
(3) L458-L511
Each figure should have its legend. You don't need to have a separate section for legend or notes. Please move them and add your legends and notes near of each figure or table.
Response: Thanks for your suggestion, we have removed figure legends and table notes at the end of the article and added supplementary materials.
Reviewer 2 Report
Manuscript of Yao Tong with co-authors ‘Cryptic Species exist in Vietnamella sinensis Hsu, 1936 (Insecta: Ephemeroptera) from Studies of Complete Mitochondrial Genomes’ represents data of three mitochondrial genomes of Vienamella sinensis. The results of the analysis according to the author indicate cryptic species within diversity of V. sinensis, in particular the sample collected in Qingyuan is significantly differ from others samples. The MS is well written, structure and formal analysis is clear. However I am not agree with the authors that this phenomenon can be called ‘cryptic species’. To be more precise I do not ruled out that the author is right, but the presented evidence are not sufficient to call it cryptic species. Data of morphology/ecology and finally nuclear genome data are needed. Indeed, acceleration rate of fixation could lead to accumulation great differences. Indirect effect of Wolbachia symbiont on mitochondrial diversity could lead to such phenomenon. And the absence of Wolbachia (or any other facultative (in respect to host) maternally-inherited bacteria) in studied samples could not be the evidence that the indirect effect can be excluded (because Wolbachia can be lost). Another way is introgression of mitochondrion heredity from close relative species. Nowadays a donor of cytoplasm probably not exist in these region or does not exist at all. So, an analysis of nuclear DNA is needed to say about cryptic species. In addition, lines 353-354 tell us that previously V. sinensis was considered as a number of species. Was it based on morphology criterion? And the last but not the least, I suppose that nobody knows about reproductive isolation between individuals of considered populations. Accumulation of genetic differences indirectly indicates on species divergence in relation to the reproductive species concept.
I think that the authors should change ‘species’ on ‘diversity’, postulate the hypothesis about cryptic species and add the program how it can be to elucidate.
Last point, in general I am ready to agree with DOI 10.1038/s41598-019-42297-5 where authors in particular stated: ‘Morphological and molecular methods should be applied in concordance to form a fine-scale multilevel taxonomic framework, and not necessarily implying only an a posteriori transformation of exclusively molecular-based ‘cryptic’ species into morphologically-defined ‘pseudocryptic’ ones.’
Author Response
Reviewer #2:
Manuscript of Yao Tong with co-authors ‘Cryptic Species exist in Vietnamella sinensis Hsu, 1936 (Insecta: Ephemeroptera) from Studies of Complete Mitochondrial Genomes’ represents data of three mitochondrial genomes of Vienamella sinensis. The results of the analysis according to the author indicate cryptic species within diversity of V. sinensis, in particular the sample collected in Qingyuan is significantly differ from others samples. The MS is well written, structure and formal analysis is clear. However, I am not agree with the authors that this phenomenon can be called ‘cryptic species’. To be more precise I do not ruled out that the author is right, but the presented evidence are not sufficient to call it cryptic species. Data of morphology/ecology and finally nuclear genome data are needed. Indeed, acceleration rate of fixation could lead to accumulation great differences. Indirect effect of Wolbachia symbiont on mitochondrial diversity could lead to such phenomenon. And the absence of Wolbachia (or any other facultative (in respect to host) maternally-inherited bacteria) in studied samples could not be the evidence that the indirect effect can be excluded (because Wolbachia can be lost). Another way is introgression of mitochondrion heredity from close relative species. Nowadays a donor of cytoplasm probably not exist in these region or does not exist at all. So, an analysis of nuclear DNA is needed to say about cryptic species. In addition, lines 353-354 tell us that previously V. sinensis was considered as a number of species. Was it based on morphology criterion? And the last but not the least, I suppose that nobody knows about reproductive isolation between individuals of considered populations. Accumulation of genetic differences indirectly indicates on species divergence in relation to the reproductive species concept.
I think that the authors should change ‘species’ on ‘diversity’, postulate the hypothesis about cryptic species and add the program how it can be to elucidate.
Last point, in general I am ready to agree with DOI 10.1038/s41598-019-42297-5 where authors in particular stated: ‘Morphological and molecular methods should be applied in concordance to form a fine-scale multilevel taxonomic framework, and not necessarily implying only an a posteriori transformation of exclusively molecular-based ‘cryptic’ species into morphologically-defined ‘pseudocryptic’ ones.’
Response: We really appreciate your comments and suggestions after reading carefully.
First of all, regarding the morphological/ecological data that you think needed to add, we appreciate but based on some previous published papers (eg: 10.1016/j.gene.2009.09.006; 10.1016/j.tig.2007.07.001. etc…), we think that our samples are considerable and important to identify the cryptic species of Vietnamella sinensis. And we will organize our morphological data into separate articles for future publications. Secondly, there is no indirect effect of Wolbachia on the cryptic specie in our proposed research samples. For proof, we conducted a verification experiment by synthesizing primers commonly used for molecular identification of the wsp gene and 16S gene of Wolbachia (Please see as below: Table 1), and tried five PCR Tm values. The results showed that Wolbachia was not found in the three different populations in this experiment (Figure 1). So it could be ruled out as indirect effect of cryptic species. In fact, we knew that Wolbachia can lead insects parthenogernesis, but we did not read any references about Wolbachia can change the mitochondrial genome sequence, which is really give us the new research direction. If it is existed, it is really very interesting! Would you like to give any references to us?
Table 1. Primers and references used for amplification.
|
Gene |
Primers |
Reference |
|
Wolbachia-wsp-81F |
5'-TGGTCCAATAAGTGATGAAGAAAC-3' |
(Zhou et al., 1998) |
|
Wolbachia-wsp-691R |
5'-AAAAATTAAACGCTACTCCA-3' |
|
|
Wolbachia-16SF |
5'-TTGTAGCCTGCTATGGTATAACT-3' |
(O’Neill et al., 1992) |
|
Wolbachia-16SR |
5'-GAATAGGTATGATTTTCATGT-3' |
Figure 1. Verification experiment of Wolbachia.
Moreover, regarding introgression of mitochondrion heredity from close relative species, the samples in this experiment do not belong to the same distribution, so the influence of introgression hybridization can be excluded. In the other words, if there is existed introgression of mitochondrion, we can be found other species with the same gene sequence. But we failed to find it. The maternal inheritance in mitochondrial genome of Insecta was be found now, no new inheritance mode is found. Although many agriculture insects were found high genetic distance, because those agriculture insects were existed in complex species in fact. But most researchers were concern on pest control not species taxonomy. In my knowledge, introgression hybridization can make the mitochondrial gene distance small not wide. In our paper, we found that the whole mitochondrial genome is different to the same species, not the same to the known species. This is the new mt genome. According to no Sympatric species of Vietnamella found in three population in this study, so we can exclude the high genetic distance in mt genome caused by introgression hybridization.
“Nowadays a donor of cytoplasm probably not exist in these region or does not exist at all. So, an analysis of nuclear DNA is needed to say about cryptic species.“ Yes, nuclear DNA is a good choice to check cryptic species. But mitochondrial genome can also a good molecular marker to identify the cryptic species.
“ In addition, lines 353-354 tell us that previously V. sinensis was considered as a number of species. Was it based on morphology criterion? ” Yes, the previously V. sinensis was considered as four different species based on morphological characters and combined into one species V. sinensis also based on morphological characters. We described the cryptic species based on mitochondrial genome. We will describe the cryptic species as new species in the future based on the molecular and morphological characters. In this paper, we want to discuss the Cryptic Species exist in Vietnamella and the relationship of Vietnamellidae.
“And the last but not the least, I suppose that nobody knows about reproductive isolation between individuals of considered populations. Accumulation of genetic differences indirectly indicates on species divergence in relation to the reproductive species concept.” Yes, it is hard to define “species”!
Reproductive isolation is a good method to check species but it is hard to check. Although “Accumulation of genetic differences indirectly indicates on species divergence in relation to the reproductive species concept”, but the high genetic difference can be checked. The species is not the aim in this paper.
For the last point, regarding your consent to "Morphological and molecular methods should be applied in concordance to form a fine-scale multilevel taxonomic framework, and not necessarily implying only an a posteriori transformation of exclusively molecular-based ‘cryptic’ species into morphologically-defined ‘pseudocryptic’ ones." Yes, it is a good suggestion. But we want to discuss the relationship and cryptic species in the paper. There have been many articles based on only mitochondrial genome data to determine the existence of cryptic species which supported this opinion.
- Feng, S.; Yang, Q.; Li, H.; Song, F.; Stejskal, V.; Opit, G.P.; Ca I, W.; Li, Z.; Shao, R. The Highly Divergent Mitochondrial Genomes Indicate That the Booklouse, Liposcelis bostrychophila (Psocoptera: Liposcelididae) Is a Cryptic Species. G3-Genes Genomes Genetics 2018, g3.300410.2017.
- Yu, D.N.; Zhang, J.Y.; Peng, L.; Zheng, R.Q.; Shao, C. Do cryptic species exist in Hoplobatrachus rugulosus? An examination using four nuclear genes, the Cyt b gene and the complete mt genome. Plos One 2015, 10, e0124825.
- Durand, J.D.; Borsa, P. Mitochondrial phylogeny of grey mullets (Acanthopterygii: Mugilidae) suggests high proportion of cryptic species. Comptes rendus - Biologies 2015, 338, 266-277.
- Yuan, Y.; Kong, L.; Li, Q. Mitogenome evidence for the existence of cryptic species in Coelomactra antiquata. Genes & Genomics 2013, 35, 693-701.
- Jia, W.; Yan, H.; Lou, Z.; Ni, X.; Littlewood, D. Mitochondrial genes and genomes support a cryptic species of tapeworm within Taenia taeniaeformis. Acta tropica 2012, 123, 154-163.
- Torricelli, G.; Carapelli, A.; Convey, P.; Nardi, F.; Boore, J.L.; Frati, F. High divergence across the whole mitochondrial genome in the "pan-Antarctic" springtail Friesea grisea: Evidence for cryptic species? Gene 2010, 449, 0-40.
- Iannelli, F.; Pesole, G.; Sordino, P.; Gissi, C. Mitogenomics reveals two cryptic species in Ciona intestinalis. Trends in Genetics 2007, 23, 419-422.
- Haruyama, N.; Naka, H.; Mochizuki, A.; Nomura, M. Mitochondrial Phylogeny of Cryptic Species of the Lacewing Chrysoperla nipponensis (Neuroptera: Chrysopidae) in Japan. Annals of the Entomological Society of America 2008, 101, 971-977.
- Vinay, T.N.; Raymond, J.A.J.; Katneni, V.K.; Aravind, R.; Balasubramanian, C.P.; Jayachandran, K.V.; Shekhar, M.S.; Vijayan, K.K. Mitochondrial DNA Study Reveals the Cryptic Species Penaeus japonicus (form-II) in Indian Waters. Journal of Coastal Research 2019, 149-155.
We also put forward the point of view that without any indirect effect on mitochondrial (mt) genome (eg. Wolbachia, bacteria...), the mt genome is accurate enough to identify cryptic species, and recently there have been many articles based on only mitochondrial genome data to determine the existence of cryptic species which supported this opinion.

Round 2
Reviewer 1 Report
No more comments